# Which Neurons Matter in IR? Applying Integrated Gradients-based Methods to Understand Cross-Encoders

## ABSTRACT

With the recent addition of Retrieval-Augmented Generation (RAG), the scope and importance of Information Retrieval (IR) has expanded. As a result, the importance of a deeper understanding of IR models also increases. However, interpretability in IR remains under-explored, especially when it comes to the models' inner mechanisms. In this paper, we explore the possibility of adapting Integrated Gradient-based methods in an IR context to identify the role of individual neurons within the model. In particular, we provide new insights into the role of what we call "relevance" neurons, as well as how they deal with unseen data. Finally, we carry out an in-depth pruning study to validate our findings.

## KEYWORDS

Information Retrieval, Cross-Encoders, Interpretability, Integrated Gradients

**ACM Reference Format:**
Anonymous Author(s). 2024. Which Neurons Matter in IR? Applying Integrated Gradients-based Methods to Understand Cross-Encoders. In *Proceedings of The 14th International Conference on the Theory of Information Retrieval (ICTIR '24)*. ACM, New York, NY, USA, 11 pages. https://doi.org/XXXXXXX.XXXXXXX

## 1 INTRODUCTION

Since the BERT [9] era, Information Retrieval (IR) has gone through many changes. This paradigm shift has seen the rise of neural IR systems in the light of the performance of BERT-base models compared to previous state-of-the-art in almost all benchmarks. This huge improvement however came at the cost of the explainability, as Transformers (and thus BERT) are extremely complex models. Despite being widely adopted, the Transformers' mechanisms remain poorly understood and this limits the explainability of neural IR models. In parallel, with the mass adoption of BERT, research on explainability and interpretability (i.e., *explainable AI*) has also seen a rapid surge. Being able to understand how models make predictions or which mechanisms they rely on not only helps with their adoption by users but also unlocks the possibility for researchers to study edge cases where they might fail, providing room for improvement. By improving our understanding of the mechanisms and signals involved when performing the IR task, we can also design new architectures or better-suited training algorithms, able to bridge actual gaps or correct misbehavior of existing systems,

and better transfer techniques, targeting more precisely domain or language-specific parts of the models.

As of today, Transformers-based models remain mostly "black boxes". Despite some successes in providing new insights about the different signals/features that neural models regard as important, their inner mechanism, i.e. how those signals are leveraged and/or combined by their different components, remains unclear. Different lines of work have emerged to address the challenge of understanding the Transformers-based models' machinery [8, 11, 15] such as probing [2], mechanistic interpretability [11] or attribution methods. Within the latter, another distinction exists between perturbation-based methods [13, 31, 32] and backpropagation-based methods [1, 33, 34, 37]. If the literature is expanding quickly in Natural Language Processing (NLP), it is scarcer in the specific domain of IR.

This paper aims at filling this gap by studying the application of a gradient-based approach, namely Integrated Gradients [37], to understand the role of neurons within a cross-encoder model, here MonoBERT [28], in an IR task. Furthermore, we hope this work will pave the way for future ones aiming at improving IR systems. We believe that understanding better how IR models work is primordial to help the field move forward and conceive new systems. As identifying these mechanisms is not trivial because of the complexity of language models, we explore the relative importance of neurons for different aspects of the IR task (notion of relevance and in-domain vs out-of-domain data). In particular, our study focuses on the following research questions:

- **RQ1** Is it possible to identify neurons involved in the classification of a passage as "relevant" (or "non-relevant") for a given query?
- **RQ2** Is it possible to distinguish neurons involved with in-domain data from those involved with out-of-domain data?
- **RQ3** How important are those neurons for the IR task?

## 2 RELATED WORKS

The IR landscape changed drastically with the arrival of Transformers [42] and BERT [9], the whole domain shifting entirely towards neural IR systems. If these models can significantly improve the quality of the retrieved content, most of them, including the most effective ones like cross-encoders [28, 29], lack interpretability and explainability. Counter-examples with the core ability to provide explanations for their predictions such as SPLADE [14] or ColBERT [18] exist thanks to their architecture which leverages a form of matching (between expanded queries' tokens and passages' tokens for SPLADE, and between contextual vectors for ColBERT). Even if some models can better explain their predictions, there is no clear understanding of the process leading to that prediction or the signals that they extracted and transformed to reach such a decision.

That is typically the reason that motivated the development of techniques able to unlock a model black box and allow researchers to look under the hood of neural networks. The different explainability techniques can be categorized into distinct families based on how they tackle the infamous "black-box" issue of neural networks.

First, probing [2] trains probe classifiers from the model's hidden representations and evaluates them on tasks associated with the primary objective for which the model was designed (e.g., for IR, such tasks can be Named-Entity Recognition, Semantic Similarity or co-reference Resolution [41, 46]), revealing the specific abilities that the model learned implicitly during its training to solve the task. However, as probing relies on external classifiers, it is considered disconnected from the original model [2]. In addition, probing methods can't be used to target specific neurons as they use each hidden representation as a whole and do not provide any explanation of the interplay between these abilities as well as their relative importance in the model's output.

Second, mechanistic interpretability [4] refers to a line of works that try to "*reverse engineer Transformers into human-understandable computer programs*" [1]. In particular, it aims at decomposing Transformers into multiple blocks whose role and relations with the rest of the model are both well-understood. This approach has provided meaningful explanations for toy models [11] or for some specific behaviors [30] but is hard to scale for an exhaustive study. For example, activation patching, or causal tracing, [25, 43] is an application of mechanistic interpretability that changes activation in some specific parts of the model observing its effect. Despite its good results in explaining the causal structure of models [12], it implies iterating over every inner output of the model, and this quickly becomes intractable. Attribution patching [38] alleviates this limitation by making use of gradient-based approximation but can lead to the apparition of false negatives [19] potentially harming its conclusions.

Finally, attribution methods aim at identifying which part of the model or the input contributed the most to a prediction. Contrary to probing, attributions are obtained directly from the model and are usually more easily scalable to study a full model, even large ones, than mechanistic interpretability. It is possible to distinguish two types of attribution methods. The first one, called *perturbation-based*, introduces perturbations of many types (masking, removing, introducing noise, etc) on the input and measures the differences in the result compared to the original output [13, 31, 32]. Perturbations have the advantage of being easily understandable while providing a good estimation of the effect of each feature on the output but are extremely costly to compute.

The second one, referred to as *gradient-based* or more generally *backpropagation-based*, recovers attributions using gradients or activations starting from the output prediction down to each layer of the model [33, 34, 37]. *Gradient-based* methods have the advantage of being faster to compute and usually have more desired theoretical properties over perturbation-based methods, such as sensitivity or additivity. Their main drawback is the difficulty of giving a human-understandable meaning to their attribution.

---

[1]Quote taken from the second thread on Transformer circuits: https://transformer-circuits.pub/

The first works in IG only consider the computation of the importance of input features. This was applied by Möller et al. [26] in the case of Siamese Encoders for the semantic similarity task. However, their work is limited to the study of tokens' attributions and mostly aims at extending IG to a setup with two inputs.

Motivated by recent studies that discovered the role of feed-forward layers within Transformers [15], several gradients-based methods were developed to provide attributions to individual neurons within the model instead of the input's features [10, 22]. In parallel, other works proposed optimizations of the computations involved in IG and its variants [22, 35], extending its possible applications and allowing to study the crucial role played by some neurons in storing knowledge [5, 7] or in specific tasks [10]. Similarly to us, Wang et al. [47] studied "skill neurons" in Transformers but through Prompt-Tuning [21]. By concatenating soft prompts $P = \{p_1, p_2, ..., p_l\}, p_i \in \mathbb{R}^d$, with $d$ the input dimension of the model, to the input, they show that some neurons have higher activation than the rest of the network and that this activation is strongly correlated to the prediction of a specific class in classification tasks. However, the theoretical ground behind the success of Prompt Tuning in identifying "skill neurons" remains unclear.

To the best of our knowledge, this work constitutes the first attempt at using Integrated Gradient-based methods, particularly the Neuron Integrated Gradients (NIG) method [35], to explain neural IR models. In contrast to Möller et al. analyzing siamese encoders for semantic signals [26], we are interested in studying the behavior of cross-encoder architecture in the IR task, especially the integration of relevance matching signals [16]. Moreover, unlike the study of skill neurons [47] we do not limit our study to the feed-forward layers in the Transformers but also include all linear transformations in the model.

## 3 NEURON INTEGRATED GRADIENTS FOR IR

In this section, we show how we leverage Neuron Integrated Gradients [35] to disentangle the neurons' importance within a cross-encoder model during the IR task. We leave the study of more costly but equally relevant alternatives as well as the extension to other architectures for future works.

### 3.1 Background

Originally designed for computing the attributions of the input features (to the output), Integrated Gradients (IG) [37] is based on path integrals. IG imply to first carefully select a baseline input $x'$ for which the model's prediction is neutral, i.e. $p(relevant) = p(non-relevant)$, and to study the straight line path $\gamma(t)$ from this baseline to the actual input $x$. Formally, $\gamma(t) = x' + t(x - x'), t \in [0, 1]$. IG are obtained by computing and integrating the gradients along the path between the baseline and the input. It has been generalized into *conductance* [10] to obtain the importance of any neuron $y$. In this framework, the importance of an individual neuron is given by the Neuron Integrated Gradients formula [35]:

$$NIG^y(x) = \int_{t=0}^{t=1} \frac{\delta F(x' + t(x - x'))}{\delta \gamma_y(t)} \frac{\delta \gamma_y(t)}{\delta t} dt \quad (1)$$

where F is a neural network, $\gamma_y(t)$ denotes the activation value of a neuron $y$ at the point $t$ of the path $\gamma$, i.e. the output $y$ of the neural network $F_y(\gamma(t))$. Note that this formulation of the attribution of a neuron has an efficient approximation based on the Riemann formulation of the integral that we leverage as described in [35].

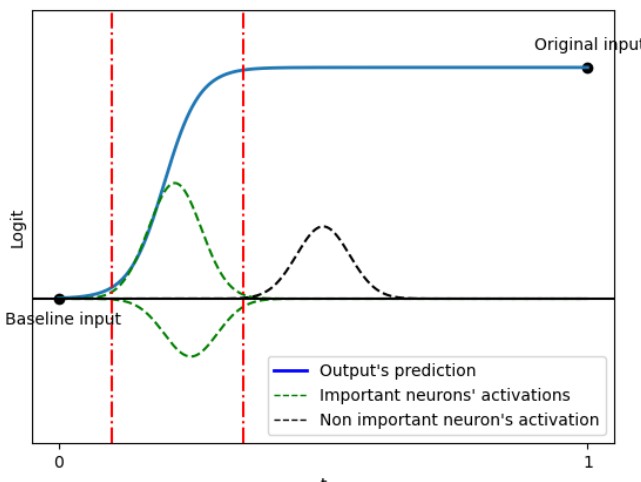

**Figure 1: Illustration of the Neuron Integrated Gradient method**

To illustrate the intuition behind Neuron Integrated Gradients, let us use Figure 1 that depicts the evolution along the path $\gamma(t)$, between the "Baseline" $x'$ and the "Original" input $x$, of the model's prediction (in blue) and the gradients of two types of neurons (black for non-important neuron and green for important neurons). The blue curve, between the "Baseline" and the "Original" input, corresponds to the value of the output logit as $t$ evolves. In the interval delimited by the two red vertical lines in the figure, a change in the input results in a significant recovery of the original output signal symbolized by a strong increase for the the blue curve. With the Neuron Integrated Gradients method, the important neurons, i.e. the neurons whose attributions with regards to the attribution will be the highest, are neurons $y$ whose gradient $\frac{\delta\gamma_y(t)}{\delta t}$ evolves at the same time as the models' output $F$. If the model's output does not change when moving the input $\gamma(t)$ along the path, then the neurons that react to this change are not important to the model's decision. Such neurons are represented in the figure by the black dashed curve: its gradient is non-zero outside of the interval where the model's prediction increases, meaning that it is not directly related to the model's decision. Conversely, the green curves correspond to neurons that are important for the output. Note that the contribution can be either positive or negative, as illustrated in the figure.

Neurons correspond to the activation of any layer/block in the model but in our work, we are interested in neurons corresponding to outputs of linear transformations (used to compute keys, queries, values, and inside the feed-forward block [42]) as their importance in many mechanisms (factual recall, performing annex tasks, etc.) has already been characterized in previous works [5, 7, 15, 41, 46, 47].

**Table 1: Comparison of different baselines' effect on the model's predictions.**

| Method | Average difference between each label |
|---|---|
| Original embedding | 1 |
| All tokens → [PAD] | **0.21** |
| Query tokens → [PAD] | 1 |
| All tokens → zero vectors [26] | 0.67 |
| Query tokens → zero vectors | 1 |

## 3.2 Adapting NIG to identify "task-related" neurons

As IR is different from other domains (CV and NLP) in which NIG [35] has already been applied, some of its aspects need to be adapted.

*Comparisons across datasets.* To make fair comparisons between attributions obtained for different datasets, we aggregate the contribution value of each linear transformation's outputs. As they are shared among tokens composing the query-document pair, we sum the conductance over tokens: a "neuron" in our experiments thus includes its corresponding outputs over all tokens. This aggregated conductance can be thought of as the importance of outputs of a linear transformation within a Transformer.

*Dependency to the input.* As pointed out by Wang et al. [47], gradient-based methods produce results that are input-dependent. As we want to identify "IR task-related" or "skills" neurons, i.e., neurons that are important for any input in the IR task, we average NIG results over multiple samples across multiple datasets and only retain the neurons with the highest mean attribution values.

*Baseline.* For images, an obvious baseline $x'$ is a black image. In IR, there is no obvious baseline, and we thus empirically verify which one is better suited as a baseline by comparing how well they degrade the relevance signal on average over 1000 inputs from the MSMARCO dataset [27]. Among the baseline that we consider, we include the method of Möller et al. [26] who study IG in the context of Siamese encoders. The authors build a baseline for which the predicted relevance probability is close to 0.5 by subtracting the embeddings of the baseline on every sample along the path between the baseline and the original input. This is equivalent to replacing every token in the input with a zero vector, i.e. an embedding where every dimension equals 0. As the authors don't try alternatives, we further compare the effect of "zeroing out" only the queries' tokens or the entire input and of simply using [PAD] tokens instead of the query or entire input tokens (the first step in Möller et al. [26] before they subtract their embeddings), on the model's predictions. We consider the values from the output of the Softmax operator. For the choice of the baseline, we subtract the value for the "non-relevant" label from the value for the "relevant" label for each of our 1000 inputs and average the difference. We compare the average difference with the original inputs and when applying our baseline obtention methods. The closer the average difference is to zero, the stronger the baseline's suppression of the original input's relevance signal. Results are in Table 1. Based on this, we decided to transform every embedding into its fully padded counterpart to obtain our baseline.

**Table 2: Details of the datasets along with their abbreviations we use in the remainder of the paper as well as the task associated with it.**

| Dataset's name | Abbreviations | Associated task |
|---|---|---|
| MSMARCO [6, 27] | ms | Web Retrieval |
| FiQA [23] | f | Question Answering |
| Natural Questions [20] | nq | Question Answering |
| BioASQ [40] | b | BioMedical-IR |
| TREC-Covid [44] | tc | BioMedical-IR |
| NFCorpus [3] | nf | BioMedical-IR |
| TREC-News [36] | tn | News Retrieval |
| TREC-Robust04 [45] | r | News Retrieval |

## 4 EXPERIMENTS

Using Neuron Integrated Gradients, we conduct several experiments using one base IR model over different datasets to compute the neuron attributions. By comparing the attributions over multiple datasets, we want to identify the core set of important neurons for the IR task (see **RQ1** and **RQ2**). To empirically verify our results, we perform a series of ablation studies where we evaluated the decrease in performance on a new set of datasets caused by the ablation of the neurons tagged as important by our attributions in the IR model (see **RQ3**).

Finally, note that the code, including all technical details, will be publicly released upon publication.

### 4.1 Experimental setup

In our experiments, we analyze the model MonoBERT [28] as it is a strong baseline and a typical cross-encoder. Despite the existence of stronger models such as MonoT5 [29], we decided to stick to MonoBERT as it is an encoder-only architecture, contrary to T5: We are interested in interpreting the model's inner mechanisms and the decoder part brings an additional layer of complexity to deal with. The version of MonoBERT we use has been fine-tuned on MSMARCO[2] [27].

Motivated by the **RQ2**, we compute attributions over several datasets, both in the same domain (ID) as the training data of our model and outside of it (OOD). For ID, we use the test set of MS-MARCO[3] from the TREC 2019 Deep learning track [6] and for OOD, datasets from BEIR [39]. We choose datasets corresponding to tasks that resemble the most a traditional IR setup in BEIR according to their classification[4]: FiQA [23], TREC-Covid [44], TREC-News [36], NFCorpus [3], BioASQ [40], Natural Questions [20]. In addition, we also include TREC-Robust04 [45] as it is also a known dataset for retrieval. Together, these datasets as well as the test set of MS-MARCO compose our attribution corpus. Table 2 summarizes the attribution datasets and their different characteristics, including their abbreviations, used in the different formulas later in the paper.

To empirically validate our findings, we further use the development set of MSMARCO (we are less focused here on the quality of the assessments because this is a ranking setup) as well as of the LoTTE benchmark [18], also spanning various domains: Lifestyle, Recreation, Science, Technology, and Writing.

### 4.2 Analysis methodology

In the case of a cross-encoder, the IR task can be seen as a series of binary classification tasks where the model has to estimate the relevance of a passage to a query. When computing NIG, we estimate separately the attributions for the "relevant" label and for the "non-relevant" label (when available [5]). We name the output of the NIG attribution method an *attribution scheme*, i.e. the set of attribution values of every neuron for either the "relevant" or "non-relevant" labels of a given dataset. We ensure before assigning one query-passage pair to the label "relevant" or "non-relevant" that the original model prediction matches the assessment. Future work could include a more fine-grained analysis by distinguishing the assessments between the unambiguous pairs and the ambiguous pairs. As we focus on understanding the generic mechanisms behind IR systems' predictions, we exclude the ambiguous pairs. For each attribution scheme, we can rank the neurons $y$ based on their mean importance $NIG^y$ and select the top $X\%$ of neurons in the model (typical values of $X \in [0.01, 0.1, 1]$). For example, we can derive the top 1% of neurons with the highest attribution value in the whole model for the "relevant" label on MSMARCO. These subsets constitute the base units of our analysis. Following previous works [26, 37] and to keep the computing time reasonable (given the number of linear transformations that we consider in our study), we approximate attributions using $N = 100$ steps. We verified this number is high enough to minimize the approximation error due to the discretization when computing the integral [24].

*Answering RQ1.* To know if it is possible to identify neurons involved in the classification of a passage as "relevant" (or "non-relevant") for a given query, we leverage the attributions for both types of labels. We start from the sets of neurons involved in the prediction of the "relevant" (positive) label and "non-relevant" (negative) label for the dataset $x$, $x \in \{ms, f, tc, tn, nf, b, r, nq\}$ (see Table 2 for the abbreviations), denoted as $P_x$ and $N_x$ respectively. These correspond to the basic attribution schemes. We suppose that the set of "core" neurons for relevance (resp. non-relevance) (**RQ1**) is the intersection of the basic attribution schemes, based on the fact that neurons specific to the IR task (for a given label) should be consistently tagged as important across datasets. We consider the relevance and non-relevance separately to replicate prior works leveraging NIG on classification tasks and who found that sets of important neurons for different labels usually do not intersect [10, 47].

*Answering RQ2.* Another important aspect of these intersections is the nature of the target domain. Indeed, to determine whether or not MonoBERT contains neurons dedicated to OOD predictions (**RQ2**), it is necessary to compare the "core" set of neurons across every dataset and the "core" set of neurons across OOD datasets only.

---

[2] The model is available on the HuggingFace' Hub: castorini/monobert-large-msmarco

[3] We compute Neuron Integrated Gradients on the test set as the development set has many false negatives

[4] The BEIR benchmark covers a total of 9 tasks, among which 3 can be considered the closest to the IR task: News Retrieval, Question-Answering (minus HotpotQA [48] which is a multi-hop QA dataset) and Bio-Medical IR

[5] BioASQ and FiQA only have relevant annotations

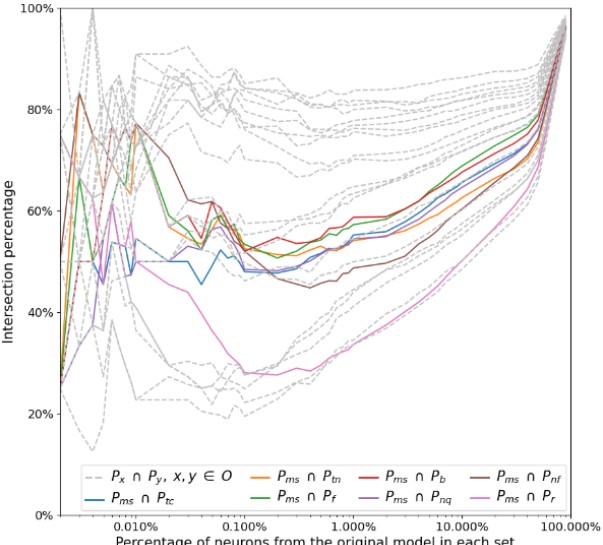

(a) "Relevant" neurons: Comparisons of the intersection between any 2 datasets in the attribution corpus. The grey dashed curves summarize the intersection between two OOD datasets.

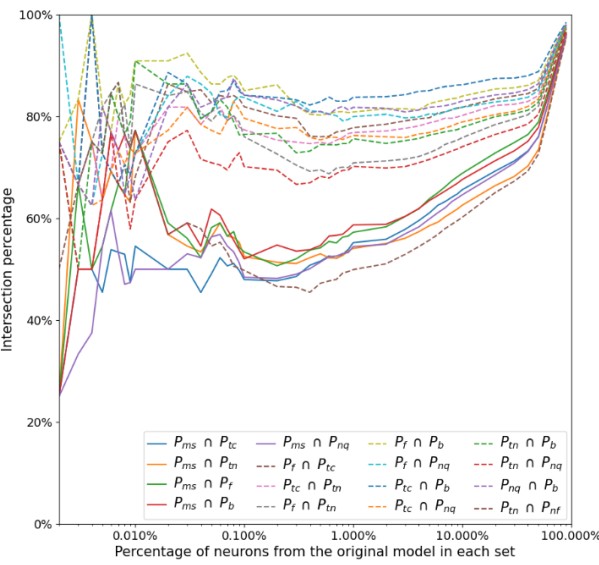

(b) Relevant neurons: Comparisons of the intersection between any 2 datasets in the attribution corpus except Robust04 and NFCorpus

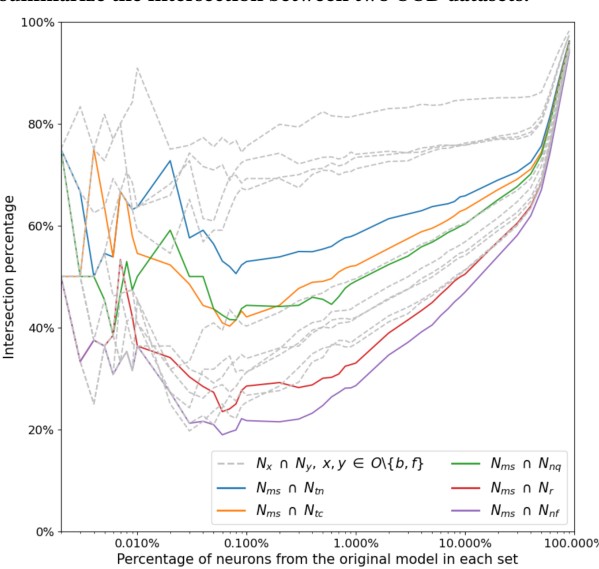

(c) Non-relevant neurons: Comparisons of the intersection between any 2 datasets in the attribution corpus except BioASQ and FiQA. The grey dashed curves summarize the intersection between two OOD datasets.

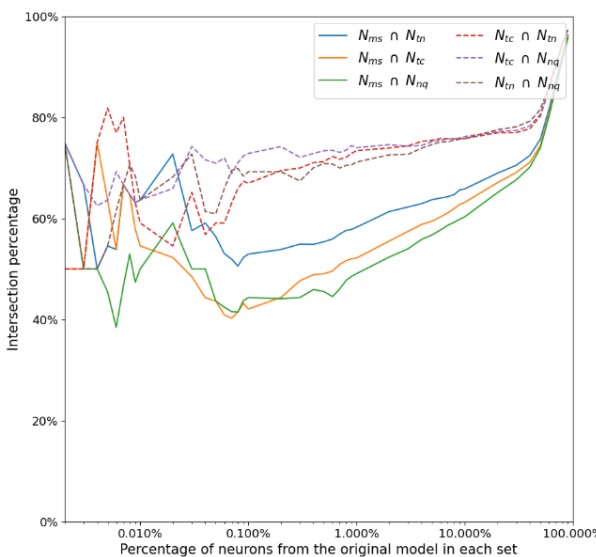

(d) Non-relevant neurons. Comparisons of the intersection between any 2 datasets in the attribution corpus except BioASQ, FiQA, Robust04 and NFCorpus

Figure 2: Percentage of intersection between pairs of attribution schemes for the label "relevant" (top) (resp. "non-relevant" (bottom)) at different percentages of pruning

*Answering RQ3.* At this point, our findings are based on the attribution from NIG but it is still unclear whether they can impact the IR task by changing the rankings produced by MonoBERT. To investigate this, we need to deprive the model of its ability to deal with relevance and/or non-relevance. Following [22], we set important neurons to zero and observe the effects on both

the model's predictions but also the IR task. As our target is to identify the most important neurons for the IR task, we do not limit ourselves to the attribution schemes composed of the top $x\%$ of the most important neurons for a single dataset and a single label. Instead, we combine attribution schemes to further refine the set of most important neurons. As intersections might not be the best

**Table 3: Summary of the notations we have introduced and their meaning. For both intersection and fusion, as long as the operation involves attribution schemes for the label "non-relevant", we exclude BioASQ and FiQA from the set datasets as they both lack "non-relevant" assessments.**

| Notation | Signification |
|---|---|
| $A = \{ms, f, tc, tn, nf, b, r, nq\}$ | The set of every dataset in the attribution corpus |
| $O = \{f, tc, tn, nf, b, r, nq\} = A \setminus \{ms\}$ | The set of every OOD dataset in the attribution corpus |
| $P_x, x \in A$ | The attribution scheme for "relevant" label on dataset $x$ |
| $N_x, x \in A \setminus \{b, f\}$ | The attribution scheme for "non-relevant" label on dataset $x$ |
| $P_x \cap P_y, x, y \in A$ | The intersection between two attribution schemes for the "relevant" label from two different datasets |
| $\bigcap_{a \in A} P_a$ | Intersection of every attribution schemes for the "relevant" label |
| $P_x \bigoplus N_x, x \in A \setminus \{b, f\}$ | Fusion of the attribution for both labels "relevant" and "non-relevant" of dataset $x$ |
| $\bigcap_{a \in A \setminus \{b, f\}} F_a$ | Intersection of the fusions of the attributions schemes "relevant" and "non-relevant" of every dataset in $A \setminus \{b, f\}$ |
| $F_A$ | Fusion of every attribution schemes together |

way to combine schemes, we explore other ways to better define the most important neurons. One obvious solution is what we call the fusion operation, where the attribution values of both relevant and non-relevant sets are averaged to compute the importance of each neuron. We denote this operation with $\bigoplus$. For any dataset x, $P_x \bigoplus N_x$ denotes the fusion of the attribution schemes for the "relevant" label and the "non-relevant" label. Please note that to ease reading, this fusion is referred to as $F_x$. Other combinations are more straightforward as they only rely on intersections between sets. Additionally, we denote the subset of all the OOD datasets as $O = \{f, tc, tn, nf, b, r, nq\}$. Similarly, we denote the set of all datasets as $A = \{ms, f, tc, tn, nf, b, r, nq\}$. Note that we do not specify the pruning level when denoting those sets.

## 5 RESULT ANALYSIS

We now comment on the results of the experiments and answer the research questions. To ease reading, we provide a summary of the notation we use in Table 3.

### 5.1 RQ1: Are there relevance-specific neurons?

Figures 2a-d. depict the intersections between the sets of relevant or non-relevant neurons, at different pruning levels. More precisely, we report for a given label:

(1) Every pairwise intersection , i.e. $P_x \cap P_y, x, y \in A$ and $N_x \cap N_y, x, y \in A \setminus \{b, f\}$ (BioASQ and FiQA lack non-relevant assessment). We distinguish the OOD/OOD datasets pairs from the MSMARCO/OOD pairs, as these will also help us answer **RQ2**;

(2) The intersection between every dataset in $O$ and $A$, i.e., $\bigcap_{o \in O} P_o$ (resp. $\bigcap_{o \in O \setminus \{f, b\}} N_o$) and $\bigcap_{a \in A} P_a$ (resp. $\bigcap_{a \in A \setminus \{f, b\}} N_a$);

(3) for any dataset $x$ in $A \setminus \{f, b\}$, we compute $P_x \cap N_x$.

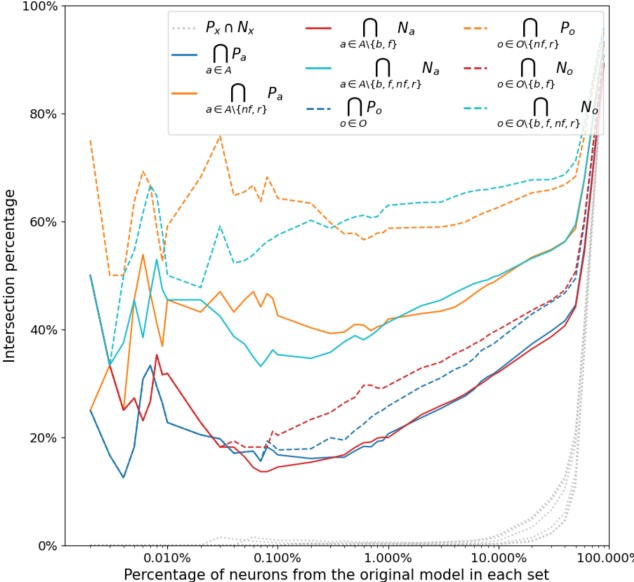

**Figure 3: Summary of the intersections amongst all the attribution schemes of every dataset (or all the datasets but Robust and NFCorpus) for a given label (either "relevant" or "non-relevant") and of the intersections between two attribution schemes (both label) for a single dataset**

In these figures, the curves describe the percentage of neurons in the intersection between 2 sets at different pruning percentages. Dashed curves correspond to the intersection between two OOD datasets and plain curves to the intersection between MSMARCO and an OOD dataset (as previously described). One can easily see that for both "relevant" and "non-relevant" predictions, there exists a set of neurons that is consistently involved across domains which means that there are neurons specifically allocated for relevance, thus answering the **RQ1**.

Furthermore, in Figure 3, which summarizes Figures 2a-d, the grey dotted lines describe the percentage of the neurons that are in common between the relevant and non-relevant attribution schemes (for the same dataset). We see that whatever the dataset $x$, $P_x$ and $N_x$ do not intersect, implying that the sets of most important neurons for each label are almost entirely different. Note that this phenomenon has previously been observed in sentiment analysis [10, 47], but we are, to the best of our knowledge, the first to report it in IR.

As the intersection between the relevant and non-relevant attribution schemes is almost empty across every dataset, it suggests the possibility to distinguish between the neurons involved in predicting the label. In addition, this observation could imply that in each case, the type of signals or mechanisms involved is different, outlining the existence of different signals in relevance (beyond semantics ones) as suggested in DRMM [16]. However, understanding exactly what are the relevance signals involved in each case would require a new set of experiments that we leave for future work.

**Table 4: % of differences on nDCG@10 between pruned and original models, following different ablation schemes. For each ablation scheme, we distinguish its impact on MSMARCO and on the LoTTE benchmark's datasets. For the latter, we average the results over the five datasets and report between parenthesis the number of datasets (out of 5) if any, for which there is a difference compared to the original model, with statistical significance ($p \leq 0.05$) under the two-tailed Student's t-test. The biggest perturbation at each percentage is in bold, the second one is underlined.**

| Sets of neurons pruned | Datasets | % of nDCG@10 diff. | | |
|---|---|---|---|---|
| | | 0.01% | 0.1% | 1% |
| $P_{ms}$ | LoTTE benchmark | -1.98 | 3.32 | 8.97 |
| | MSMARCO dev set | 3.11 | 4.87 | 8.94 |
| $N_{ms}$ | LoTTE benchmark | **2.15** | **35.55** (5) | **48.40** (5) |
| | MSMARCO dev set | 2.43 | **42.92** (1) | **52.09** (1) |
| $\cap_{o \in O} P_o$ | LoTTE benchmark | -0.11 | 0.44 | -0.44 |
| | MSMARCO dev set | 0.24 | 0.50 | 3.54 |
| $\cap_{o \in O \setminus \{b,f\}} N_o$ | LoTTE benchmark | 2.11 | 3.29 | 8.41 |
| | MSMARCO dev set | **3.95** | 2.38 | 10.17 |
| $\cap_{a \in A} P_a$ | LoTTE benchmark | -0.11 | -0.01 | -0.31 |
| | MSMARCO dev set | 0.24 | 0.45 | 3.69 |
| $\cap_{a \in A \setminus \{b,f\}} N_a$ | LoTTE benchmark | 2.11 | 3.25 | 8.71 |
| | MSMARCO dev set | **3.95** | 2.44 | 12.24 |
| $F_{ms}$ | LoTTE benchmark | -2.22 | 17.96 (2) | 33.71 (4) |
| | MSMARCO dev set | -0.62 | 29.32 (1) | 37.43 (1) |
| $\cap_{o \in O} F_o$ | LoTTE benchmark | 0.00 | 0.66 | -1.48 |
| | MSMARCO dev set | 0.00 | 0.44 | -1.46 |
| $\cap_{a \in A} F_a$ | LoTTE benchmark | 0.00 | 0.66 | -1.49 |
| | MSMARCO dev set | 0.00 | 0.44 | -1.38 |
| $F_A$ | LoTTE benchmark | -0.74 | -2.33 | 9.75 (1) |
| | MSMARCO dev set | 0.09 | 2.29 | 11.15 |
| **Random baseline** | LoTTE benchmark | 0.00 | -0.53 | -0.93 |
| | MSMARCO dev set | 0.00 | 0.09 | -0.12 |

## 5.2 RQ2: Do neurons for ID predictions differ from those for OOD predictions?

Another interesting observation that we can draw from Figures 2a and 2c is that, if we consider every possible dataset for each label, there does not seem to be a clear distinction between the neurons involved only with OOD datasets and with MSMARCO and an OOD dataset. However, if we are more careful and analyze further the impact of each dataset, we note in Figures 2b and 2d that if we leave aside NFCorpus and Robust, the plain curves are now completely separated from the dashed ones. This means that the intersections between two OOD datasets have a higher number of neurons than between MSMARCO and an OOD dataset. This observation is further confirmed in Figure 3. In this figure, each pair of curves with the same color represents the intersection between all the OOD datasets and every dataset in the attribution corpus for both labels, modulo NFCorpus and Robust (see the red and purple pairs). One can easily verify that in every case, even when including NFCorpus and Robust in the set of datasets, the OOD datasets have a higher percentage of intersections together than when we add MSMARCO to the mix. Figure 3 already showcases the existence

of two sets of neurons consistently involved in the prediction of either "relevant" or "non-relevant" labels across domains. It further suggests the existence of two additional sets of neurons, completely different from the first two sets, dedicated to OOD predictions.

As an additional and distinct set of neurons is involved consistently, it seems as if predictions outside of the training domain of the model are somehow handled differently. This observation (if consistent across models) motivates future works to better understand the role of this specific set of neurons when dealing with OOD data and to design better adaptation methods for IR systems.

## 5.3 RQ3: Can NIG identify neurons important for the IR task?

To explore the impact of our observations in an IR setup, we conduct an ablation study using the corpus described in Section 4. For each dataset, we select a subset of queries and associate each relevant passage with 20 others retrieved by BM25. The ablations are conducted following different ablation schemes, coming either directly from the attribution schemes of each dataset or by combining some of them. As a baseline to our ablations, we use a random ablation

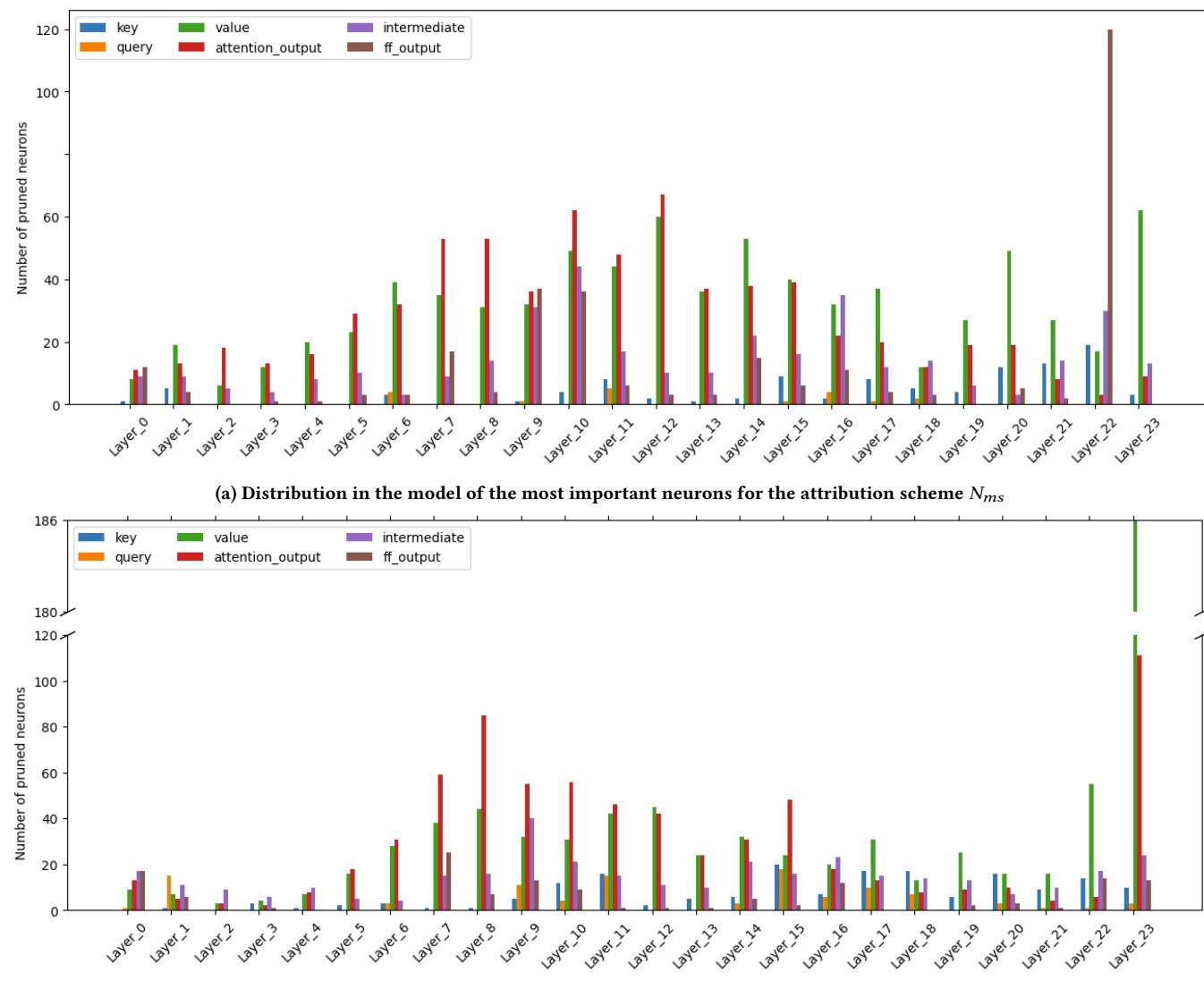

(a) Distribution in the model of the most important neurons for the attribution scheme $N_{ms}$

(b) Distribution in the model of the most important neurons in for the attribution scheme $F_{ms}$

Figure 5: Number of important neurons at each layer ( 1% pruning).

scheme, where the same amount of neurons and in the same layers that a given attribution scheme are selected. To account for randomness, we averaged the results over 50 repetitions, each time removing a different set of neurons. For each query, we measure the difference in nDCG@10 between the original MonoBERT model and its pruned counterparts when re-ranking the list of passages. For both the random baseline and the different attribution schemes, we consider three levels of pruning: 0.01%, 0.1%, and 1% and report the average differences in nDCG@10 in Table 4.

*Description of the attribution schemes applied to the original model.* Inspired by the previous experiments, we first compute the results obtained when pruning following the attribution schemes based on MSMARCO, $P_{ms}$ and $N_{ms}$, and on the intersection of the OOD datasets ("relevant" and "non-relevant" separately at first),

i.e. $\bigcap_{o \in O} P_o$ and $\bigcap_{o \in O \setminus \{b,f\}} N_o$, and every dataset, i.e. $\bigcap_{a \in A} P_a$ (resp. $\bigcap_{a \in A} N_{A \setminus \{b,f\}}$). Finally, as the IR task involves both types of relevance signals at the same time, we also combine "relevant" and "non-relevant" attribution schemes together by merging $P_{ms}$ and $N_{ms}$ as $P_{ms} \bigoplus N_{ms} = F_{ms}$. In addition, we also consider the intersections of the fusion schemes $F_x$ together such as $\bigcap_{o \in O} F_o$ and $\bigcap_{a \in A} F_a$ ( for each dataset, we first compute the set of neurons using fusion, before doing the intersection over the datasets). Last, we compute the global fusion of all the original schemes together, simply denoted as $F_A$.

From Table 4, we first observe that the random pruning baseline does not significantly impact the performances of the model. When pruning only 0.01% of the neurons, the performances are not altered at all. Higher levels of pruning (0.1% and 1%) produce changes of

at most 2% on a single dataset. When it comes to the attribution schemes obtained from our combinations, we note that even if it is not statistically significant, some of them already impact negatively the IR metrics when pruning as little as 0.01% of the neurons (around 20 neurons before any intersection). For higher levels of pruning, we observe larger degradation which eventually becomes statistically significant on some datasets. As an answer to **RQ3**, it seems that NIG can identify neurons that matter for the IR task as removing them negatively impacts IR metrics. Another observation is that one of the best attribution schemes is $F_{ms}$. This highlights the value of fusion in selecting relevance-sensitive neurons but also the importance of considering both relevant and non-relevant attributions. Altogether, this shows the possibility to identify neurons important for the IR task with NIG.

Beyond the scope of **RQ3**, Table 4 further helps us to understand the relations between these different sets of neurons and the IR task. In particular, perturbing the original model using "non-relevant" attribution schemes have more impact on performance compared to their "relevant" counterpart (i.e. $N_{ms}$ has more impact than $P_{ms}$, and likewise for intersections of attribution schemes).

## 6 DISCUSSIONS

Beyond the scope of the research questions, attributions from Neuron Integrated Gradients offer multiple new insights into the inner mechanisms of MonoBERT. In particular, Figures 4a and 4b give more details on the distribution of important neurons across the model layers and components for the two schemes with the biggest impact at 1% of pruning percentage, i.e. $N_{ms}$ and $F_{ms}$. From these figures, we observe that a significant peak in the number of important neurons occurs around the last two layers. We suspect this peak is associated with the concentration of all the signals into the [CLS] token as the model's output uses CLS-pooling. Even if it is smaller in magnitude, both figures contain a second peak located around the middle layers. This peak of activation is spread across layers 7 to 12 and emphasizes the role of these mid-level layers in the model's predictions. Interestingly, the position of this peak in the model matches the conclusions of other studies based on probing which show the importance of intermediate layers' representations in IR [46].

In addition, when looking at the details of which transformations have the most important neurons in these layers, we remark 1) the omnipresence of the last linear transformation in the attention mechanism ("attention_output") and of the value [42] and 2), the absence of important neurons in the key and query's linear transformations. If, as we suspect, these mid-level layers are involved in relevance matching (semantic and lexical), we interpret these 2 observations as matches occurring at the level of the query and key matrices before being filtered by the value and propagated to the upper parts of the model. When computing attributions, neurons appear more important in the value matrices because these are responsible for the filtering – signals in key and matrices can be considered redundant. Following this matching process, we note that the relative importance of feedforward layers also increases in mid-level layers ("intermediate" and "ff_output"), which could mean that they are used to integrate this information.

*Impact of the baseline's choice.* As detailed in Section 3.2, we carefully select our baseline to compute NIG attributions by empirically validating that it erased most of the relevance signal in the original input embeddings. This design choice is crucial as we observed different behaviors when running through the experimental process with the baseline proposed by Möller et al. [26]. Even if it did not change the conclusions, both the ablations and the observations were partially impacted: the biggest differences in Table 4 were less pronounced. For instance, for the in-model distribution of the important neurons, the peak in the middle layers was even more important, contrary to the peak in the last two layers. This reminds us that NIG attributions are dependent on the choice of a good baseline that can otherwise hinder results and conclusions.

## 7 CONCLUSION

In this paper, we present an adaptation of the Neuron Integrated Gradients attribution method that fits with the IR task, applied to MonoBERT. Our analyses highlight that within the model, it is possible to identify neurons specifically allocated to determine the relevance of a passage to a query. By extending our study across multiple datasets, we have been able to identify a core set of neurons related to the notion of "relevance" and have demonstrated the existence of a different set of neurons important in the case of OOD data. Finally, we empirically demonstrate that neurons identified by NIG are actually related to the IR task by performing multiple ablations. Overall, our study shows that the relevance of a passage is treated by two independent sets of neurons that do not depend on the dataset. Our statements link one set of neurons to matching signals particularly and another one to domain adaptation.

This work is not without limitations and could benefit from exploring additional neural IR architectures/models. The number of relevance judgments (particularly negatives) also might hinder our conclusions. Having this in mind, we however believe that our analysis provides interesting outcomes regarding the nature of neurons in the IR task and paves the way towards the design of more robust and generalizable neural IR models.

*Future works.* Our work paves the way for many follow-ups to refine the observations that we make on the role of some particular neurons in MonoBERT in the IR task. In particular, it would be interesting to explore methods such as those inspired from mechanistic interpretability, that are more costly, on the reduced scope of layers or blocks in the model that we have identified as more relatively important than the others or the role of the core set of OOD neurons. To expand our work, other models could also be considered, either stronger cross-encoders such as MonoT5 [29], bi-encoders [17] (extending the work from Möller et al. [26]) or more recent architectures such as ColBERT [18] or SPLADE [14].

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
