# OpenReview forum: "Which Neurons Matter in IR? Applying Integrated Gradients-based Methods to Understand Cross-Encoders"
_ACM.org/SIGIR/ICTIR/2024/Conference — ICTIR 2024_

### Official Review · Reviewer_tfhS · 2024-05-13

**Rating:** 1
**Confidence:** 3

**Objective Part Of Review:**

The authors leverage the well-established "Integrated Gradients" technique to investigate which neurons are important for ranking with a cross-encoder. The proposed methodology is sound. Empirically, the method shows that the MonoBERT model has a set of neurons that is important to the IR task, since its pruning hurts IR performance wrt to NDCG when compared to random pruning. The evaluation was comprehensive and done on both in-domain data (MS-MARCO) and/or out-of-domain (other BEIR datasets).


Concerns/Comments:
- The study was conducted on a single architecture (MonoBERT). Generalization of the findings to different architectures/base models is not addressed, although it is important. For instance, a similar study conducted with MonoT5 could provide better insights into the behaviors regarding the [CLS] token hypothesized in section 6 (since MonoT5 does not make use of this token, similar behavior could contradict the hypothesis).

- The authors identify the low number of negatives used as a potential limitation (20 BM25 negatives per query). Besides that, it would also be interesting to see the behavior of the method when we change the type of negatives. Do we have the same set of "important neurons" when dealing with easy (e.g., random) and hard(er) negatives?

- In Table 3, the authors evaluate on a subset of the queries, and 20 negatives per query. Why? The full set of queries and a deeper re-ranking threshold could provide a stronger claim (e.g., MS-MARCO dev top-100 BM25 re-ranking with x% random neurons pruned vs x% "important neurons" pruned)

**Subjective Part Of Review:**

The problem is interesting, and, overall, the paper is easy to follow. The introduction to the Integrated Gradients method, which is the core of the work, was sufficient. Figures 2 and 3 were hard to understand, mostly because the notation for the datasets was a bit confusing. The results are interesting, particularly the "important neuron" distribution among layers and modules.

---

### Official Review · Reviewer_F6YR · 2024-05-18

**Rating:** 2
**Confidence:** 4

**Objective Part Of Review:**

This paper applies the approach of Neuron Integrated Gradients (NIG) to examine the role and importance of neurons in a MonoBERT network used in an IR task. NIG has been applied to NLP and CV, but this is claimed to be its first application to IR. The paper states three clear Research Questions to do with identifying neurons involved in classification as relevant or non-relevant, neurons involved with in-domain and out-of-domain data, and finally the importance of neurons. Experiments are done on eight publicly available datasets. The results of the experiments are discussed in detail. The results show empirically that different sets of neurons are involved with predicting the relevant and non-relevant label, and that predictions for out-of-domain-data are handled differently. Extensive ablations are done to answer the third research question. All of the work opens up further questions for future work. Limitations are discussed at the end.

Minor comments:
- I don't quite understand what is meant by "attribution corpus".
- There is no Figure 4, the numbering of figures jumps from 3 to 5

**Subjective Part Of Review:**

The results are interesting - it is nice to see a paper examining the internals of how a neural network works rather than aiming to be top of a leaderboard.

As the paper opens up more questions based on its results, it could be the basis for much future work.

---

### Official Review · Reviewer_YzNF · 2024-05-19

**Rating:** 1
**Confidence:** 5

**Objective Part Of Review:**

The paper applies the well known technique of Integrated Gradients to probe the MonoBERT ranking model. With this parameter attribution technique, the authors find that there are subsets of parameters (neurons) that are responsible for modeling relevance.
The findings are quite intuitive because relevance modeling is essentially a pairwise classification task with negative examples. The idea of comparing the parameter weights across datasets doesn't seem convincing because these values would no way be comparable.

**Subjective Part Of Review:**

The paper is relatively well written. The problem is an interesting one. I find the idea of comparing across datasets too far-stretched.

I don't think this paper will be a good fit for ICTIR, mainly because there's little theory in the work. Nonetheless, the paper presents some solid analysis with existing explainability methods on a novel task.

The paper needs to use more readable notations (or even simple names) in the tables and figures, e.g., all the notations in the legends of the plots and Table 4 should be precisely defined with meaningful names.

Finally, there has been lots of work on term match attribution in IR, which I think is more important for the ranking task. The authors needs to cite those papers, e.g.,

Jonas Wallat, Fabian Beringer, Abhijit Anand, Avishek Anand: Probing BERT for Ranking Abilities. ECIR (2) 2023: 255-273
Maria Heuss, Maarten de Rijke, Avishek Anand: RankingSHAP - Listwise Feature Attribution Explanations for Ranking Models. CoRR abs/2403.16085 (2024)
Avishek Anand, Lijun Lyu, Maximilian Idahl, Yumeng Wang, Jonas Wallat, Zijian Zhang: Explainable Information Retrieval: A Survey. CoRR abs/2211.02405 (2022)
Manisha Verma, Debasis Ganguly: LIRME: Locally Interpretable Ranking Model Explanation. SIGIR 2019: 1281-1284
Procheta Sen, Debasis Ganguly, Manisha Verma, Gareth J. F. Jones: The Curious Case of IR Explainability: Explaining Document Scores within and across Ranking Models. SIGIR 2020: 2069-2072

---

### Meta-Review · Area_Chair_oS2a · 2024-05-30

**Recommendation:** Accept (Oral)
**Confidence:** 3

**Metareview:**

The submission adapts the method of Integrated Gradients to a transformer-based model for IR, namely MonoBERT. With this adaption, it becomes possible to inspect the model to see which neurons in the network are responsible for capturing relevance. To demonstrate the potential usefulness of the developed approach, the paper includes extensive experiments conducted on eight publicly available document collections. An interesting finding is that there appears to be a core set of neurons that seems responsible for identifying relevance regardless of the document collection that the model was trained on.

All reviewers agreed that the submission tackles an interesting and important problem. Adding to this, the submission was considered to be generally well presented. However, the reviewers also pointed out some points that the authors should take into account. This includes minor presentation issues such as the dangling references to non-existing figures. More severely, the submission should invest more effort into discussing prior research as pointed out as mentioned in the reviews. The authors are encouraged to address the points raised by the reviewers.